# Warming-induced contraction of tropical convection delays and reduces tropical cyclone formation

### Gan Zhang [1]✉

The future risk of tropical cyclones (TCs) strongly depends on changes in TC frequency, but models have persistently produced contrasting projections. A satisfactory explanation of the projected changes also remains elusive. Here we show a warming-induced contraction of tropical convection delays and reduces TC formation. This contraction manifests as stronger equatorial convection and weaker off-equatorial convection. It has been robustly projected by climate models, particularly in the northern hemisphere. This contraction shortens TC seasons by delaying the poleward migration of the intertropical convergence zone. At seasonal peaks of TC activity, the equatorial and off-equatorial components of this contraction are associated with TC-hindering environmental changes. Finally, the convection contraction and associated warming patterns can partly explain the ensemble spread in projecting future TC frequency. This study highlights the role of convection contraction and provides motivation for coordinated research to solidify our confidence in future TC risk projections.

As anthropogenic warming rises sea levels, increases extreme precipitation, and strengthens the most intense storms[1,2], the future global risk of tropical cyclones (TCs) strongly depends on the changes in TC frequency[3]. Despite extensive studies over the past two decades, the projected changes in global TC frequency remain uncertain[1–3]. While most global climate models project a decrease in global TC frequency, a statistical-dynamical downscaling model[4] that assumes unchanged statistics of convective disturbances suggests an increase in global TC frequency[1]. The direction of projected changes also depends on model parameters[5,6] and input variables[7]. For example, a set of high-resolution (25-km) climate simulations suggest that future TC frequency increases[5] as surface warming generates more seeding convective disturbances[6]. The diverging projections limit confidence in scientific understanding and complicate climate risk management.

The slow progress in reconciling the projection differences is increasingly attributed to the lack of a theoretical framework that fully explains the simulated changes and helps falsify alternative model assumptions[1,3]. Most theories recognize the essential role of atmospheric convection in TC development and are formulated around the

variables related to convective processes. One line of thought is framed around the warming-induced changes of the vertical mass flux[8]. It suggests that TC genesis is hindered by a reduction of time-mean upward mass flux[9–11] and potentially by an increase of upward mass flux per TC[12]. Another line of research emphasizes the increasing saturation deficit of a warmer troposphere. It suggests that the drying suppresses TC development by slowing the saturation of the convection environment[13,14] and facilitating the ventilation of TC convection[15,16]. Lastly, an emerging research line suggests that TC frequency may increase due to ocean warming, which promotes convective activity (including TC-seeding disturbances[6]) and expands the latitudinal range of the TC-permitting environment[17,18]. Though all developed around convection changes, these research lines rarely intersect with each other, and their connections with large-scale climate changes have not been well explored.

When explaining the climate impacts on TC activity, another line of active research builds on the concept of the Hadley circulation[19–22]. The Hadley circulation is a global-scale overturning circulation characterized by equatorial ascent and subtropical descent. A key feature

[1]Department of Atmospheric Sciences, University of Illinois at Urbana-Champaign, 1301 W. Green Street, Urbana, IL 61801, USA.
✉e-mail: gzhang13@illinois.edu

of the Hadley circulation is its concentrated equatorial convection, or the intertropical convergence zone (ITCZ). The ITCZ is at the center of global energy transport and embodies rich interactions with other weather-climate processes[23]. They collectively modulate the large-scale environment and dynamical pathways of TC development. Under anthropogenic warming, climate models and theoretical arguments suggest the Hadley circulation will weaken and expand poleward. Although these circulation changes can be complicated by natural variability[24], the projections have been adopted to explain the observed and simulated changes in TC activity in qualitative and hypothetical ways[20,22]. In the current climate regime, the configuration of the Hadley circulation is dictated by the location of tropical convection[25,26], with the equatorial convection disfavoring TC development and the off-equatorial convection favoring TC development[27]. Despite the intense interest in the Hadley circulation, past TC-climate studies paid limited attention to the seasonal migration and future changes of tropical convection.

In this work, we analyze high-resolution large ensemble climate simulations and show that a warming-induced contraction of tropical convection delays and reduces TC formation. We start with examining the seasonal associations between tropical convection and TC activity. In the simulated future climate, the warming-induced contraction involves equatorial strengthening and off-equatorial weakening of tropical convection. We link these changes to the shortening and delaying of TC seasons, as well as suppressed peak-season activity. Finally, we show the convection contraction and associated warming patterns partly explain the uncertainty in the projections of future TC frequency.

## Results

### Tropical convection and TC activity

While the annual means of TC activity and tropical convection do not share the same latitudes, their seasonal cycles suggest a close association (Fig. 1a). The seasonal migration of tropical convection generally follows the insolation and the maximum of tropospheric moisture content (or moist static energy). During the equinoctial seasons, the convection concentrates near the equator and contributes to diverging upper-level outflows that descend in both hemispheres (Supplementary Fig. 1a). In this TC-hindering circulation regime, the poleward flanks of the ITCZ are characterized by mid-tropospheric dryness related to descending motion and strong vertical wind shear related to subtropical westerlies. As the ITCZ migrates away from the equator during solstitial seasons, the convection in the summer hemisphere intensifies and drives an intense overturning cell with air ascending in the summer hemisphere and descending in the winter hemisphere (Supplementary Fig. 1c). Although mid-tropospheric dryness and high wind shear dominate the winter hemisphere, wet and low-shear conditions prevail in the summer hemisphere and favor TC development. Consequently, the spatial-temporal distribution of tropical convection effectively regulates the TC activity (Fig. 1a). Beyond the seasonal forcing, this sensitivity of TC activity to the convection location holds for a wide range of perturbations around the current climate[18,27,28].

Under anthropogenic warming, an analysis of recent climate simulations[29,30] and past studies[31–35] suggest that tropical convection will contract toward the equator. In a large ensemble of TC-permitting climate simulations (Methods), this contraction manifests as the strengthening of equatorial convection at the cost of the off-equatorial convection on the poleward flanks of the ITCZ (Fig. 1b). The contraction is evident through seasons and is the most robust around the transition from the boreal spring to boreal summer. In the early part of TC seasons, this contraction delays the poleward migration of the ITCZ and hinders the circulation shift towards the TC-favoring state. The contraction is also evident in the simulations of Coupled Model Intercomparison Project Phase 6 (CMIP6)[30] (Supplementary Fig. 2). Despite some inconsistency in the southern hemisphere, the

CMIP6 simulations concur with the large ensemble simulation and suggest the contraction of tropical convection is a robust response to anthropogenic warming.

In the previous CMIP simulations, similar signals of the convection contraction have been identified in the annual mean[31,32] and the seasonal cycle[33–35]. This warming-induced contraction likely arises from an enhanced equatorial warming[36] and cloud radiative feedback[32,37], which sharpen the meridional gradients in moist static energy and the convection activity[33,35,38]. When viewed as a delay and an equatorward displacement of the seasonal migration of tropical convection, this contraction can also be interpreted as a response to the warming-induced enhancement of cross-equator energy transport from the autumn hemisphere to the spring hemisphere[34,39]. The projected contraction depends on the equatorial warming[35], as well as model details of moist static energy and vertical velocity profiles[40].

The robustly projected contraction of tropical convection has so far escaped the scrutiny of many TC-climate studies. The existing studies mostly examine the ITCZ around the peak TC seasons and focus on the intensity and latitude of the precipitation maximum. Yet the contraction of tropical convection has no apparent impact on the latitude of precipitation maximum at the peak TC seasons (Fig. 1b). Instead, the contraction displaces the global ITCZ the most in the early part of TC seasons. Meanwhile, existing studies focus on the precipitation maximum, making it hard to detect subtle changes in the latitudinal distribution of tropical convection (e.g., equatorial convection increases). Lastly, the total precipitation increases poleward of the ITCZ due to the warming-related increase of atmosphere vapor content and deepening of the troposphere[41]. For many precipitation-based metrics, this widespread wetting likely blurs the signal of the convection contraction (Supplementary Fig. 3).

The contraction of tropical convection is accompanied by a statistically significant decrease in the frequency of TC genesis (Fig. 1c). This frequency decrease is qualitatively consistent with the impacts of stronger equatorial convection and weaker off-equatorial convection. Interestingly, the simulated TC decrease is season-dependent and mostly appears near the ITCZ. In the transition months of TC seasons (e.g., May–Jun in the northern hemisphere), the TC frequency decrease is relatively small in absolute values but represents a large fractional decrease. This feature suggests a possible shortening of the TC seasons associated with the delay of convection migration. Near the peak of TC seasons (e.g., Aug–Sep in the northern hemisphere), the TC frequency decrease is large in magnitude and preferentially occurs close to the ITCZ. This proximity suggests the TC frequency decrease is tied to the convection contraction. Motivated by the season-dependent equatorial and off-equatorial changes, the ensuing discussion will examine the shortening of the TC seasons and the peak-season changes in detail.

### Shorter and delayed TC seasons

Despite some biases in the simulated TC seasons (Supplementary Fig. 4), the large ensemble simulation suggests that anthropogenic warming shortens and delays the TC seasons in individual basins (Fig. 2). The probability distributions of TC genesis in the historical and warming simulations show differences with magnitudes up to ~0.15. Consistent with Fig. 1c, individual basins often experience a more pronounced suppression of TC activity in the transition seasons. If one defines the TC seasons with a 0.01-probability threshold of activity, the warming induced changes shorten the TC seasons by approximately 5–10%. Alternatively, the annual cycle can be divided into three-month periods consisting of a peak, a quiet, and two transitional seasons. Supplementary Fig. 4 shows that the transition periods account for ~13% (North Atlantic) to ~200% (East Pacific) of the projected decreases in TC frequency, with the six-basin average being ~75%. This suggests the importance of transition periods despite some differences between basins.

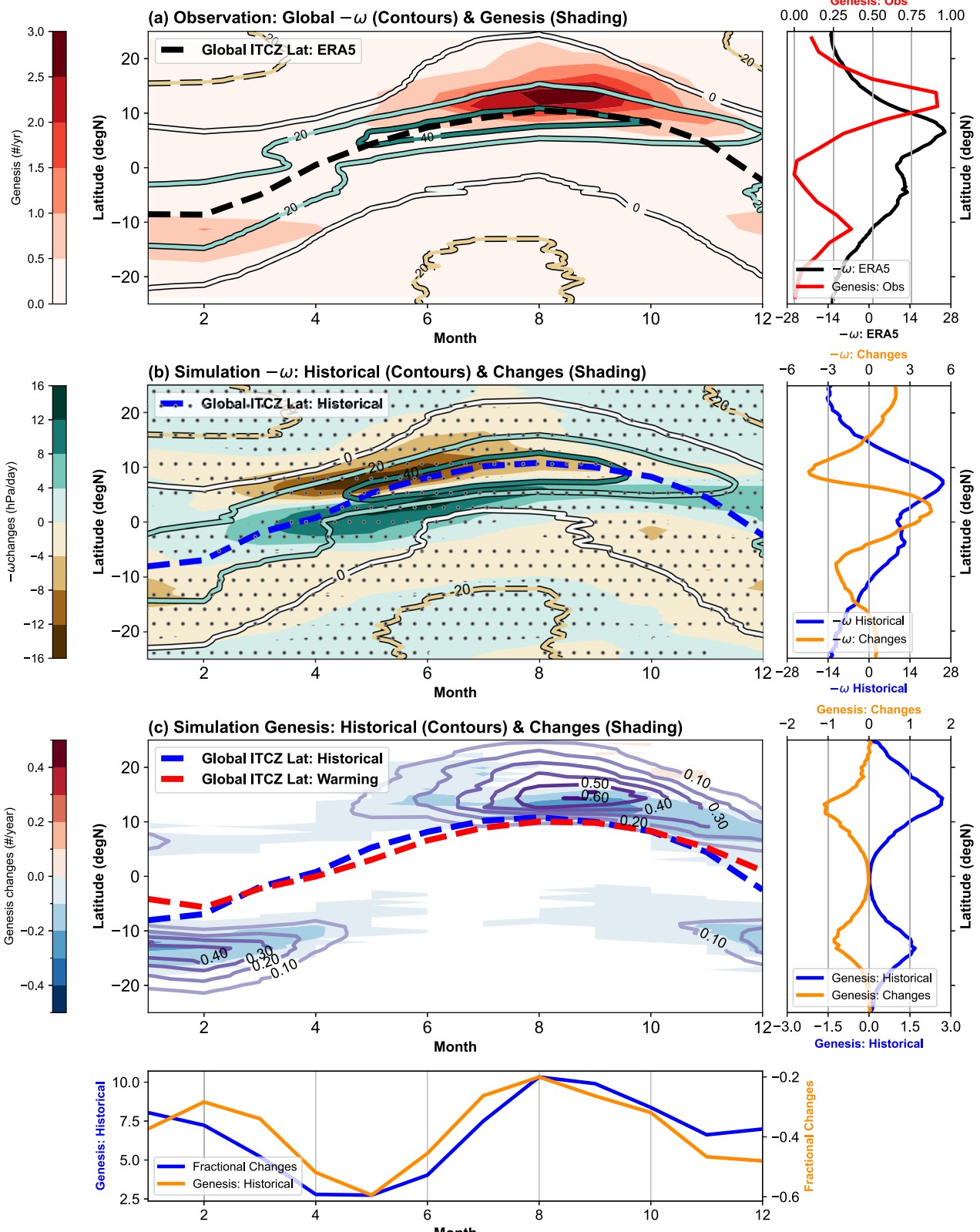

Across the individual basins, one of the most consistent signals is the delay of the early part of TC seasons, which are around June for the northern-hemisphere basins and December for the southern-hemisphere basins. With the 4-K warming, the delay is fewer than 15 days in all the basins except for the East Pacific. Other less robust changes in the simulated seasonal cycle of TC activity include an enhanced concentration of TC genesis near the season peaks and a minor seasonal delay in the average time of TC genesis. For example, such seasonality changes are pronounced in the Northern Indian basin and alter the basin's bimodal distribution of TC geneses. The changes in this basin are consistent with Murakami et al.[42] and the regional monsoon changes[43], though some quantitative aspects are likely model-dependent[44].

The simulated shortening and delay of the TC seasons are consistent with the responses to warming-induced convection

**Fig. 1 | The observed and simulated monthly evolution of tropical convection and tropical cyclone (TC) genesis frequency. a** The 500-hPa vertical motion (-$\omega$) (hPa day$^{-1}$; contours) from the reanalysis and the TC genesis frequency (yr$^{-1}$; red shading) from the best track dataset. The analysis uses -$\omega$ since negative $\omega$ values correspond to the upward motion. **b** The vertical motion in historical simulation (contour) and its responses to the 4-K warming (warming – historical; shading) from the large ensemble simulations (Methods). The changes at the 99%-confidence level are marked with stippling. **c** The TC genesis frequency in historical simulation (contour) and its response to the 4-K anthropogenic warming (warming – historical; shading). The signals below the 99%-confidence are masked out. In (**a**, **b**, **c**), the mean latitudes of the weighted Intertropical Convergence Zone (ITCZ) (Method) of the reanalysis data (1981–2010), historical simulation (HPB), and warming simulation (4-K) are shown as black, blue, and red dashed lines, respectively. The vertical motion is zonal means, and the TC genesis frequency is zonal sums. The annual means (right) and the meridional aggregations (bottom) of these quantities are displayed in auxiliary side panels. These quantities are denoted with colored axis labels. Source data are provided as a Source Data file.

contraction, but the observed and modeled changes in the TC season length can be complicated by other factors. For example, observational studies suggest the Atlantic TC season became longer[45] and started earlier[46] in recent decades, consistent with the concurrent increase in the number of Atlantic hurricanes and a large-scale environment more conducive for genesis[47]. It is possible that either natural variability[47] or anthropogenic forcings other than greenhouse gases (e.g., aerosols[48]) dominated the observed changes in an individual basin, especially as the current anthropogenic warming is substantially weaker than the 4-K warming. Another complication arises from a model that projects an increase in TC frequency[4]. When driven by the large-scale changes in the CMIP5 simulations, Emanuel's statistical-dynamic downscaling suggests the TC seasons become longer in most basins, in contrast to the results of climate models driven by the same CMIP model changes. A recent examination of Emanuel's statistical-dynamical downscaling suggests that it tends to miss early- and late-season storms[49], implying some potential weakness representing the length of TC seasons. Meanwhile, the seasonal cycle in the large ensemble simulation is not perfect either (Supplementary Fig. 4). A detailed assessment of the differences between our large-ensemble simulations and Emanuel's statistical-downscaling results is left for future research.

**Suppressed peak-season activity**

We next explore why TCs decrease near the ITCZ at the peak of TC seasons, as well as how it is related to the contraction of tropical convection. Recently, Hsieh et al.[50,51] proposed that variations in the TC frequency can be conceptually interpreted as a product of changes in the frequency of rotating convective disturbances ("seeds") and the probability of disturbance-to-TC transitioning[50–52]. Hsieh et al.[50,51] suggested that the former is correlated with the seed propensity index (SPI), and the latter inversely scales with the vertical wind shear and tropospheric dryness. We assume the SPI scales with the frequency of rotating convective disturbances[50,51,53] in the large ensemble simulation (Methods) and examine the latitudinal changes of the SPI, wind shear, and mid-tropospheric humidity (Methods) at the peaks of TC seasons. Given the active debate about the definition of TC "seeds"[50,53] and their role in the TC climatology[49], the upcoming discussion of the SPI only serves as an example of linking existing hypotheses about future TC changes to the contraction of tropical convection.

Consistent with the peak-season decrease of TC frequency, the SPI decreases near the northern-hemisphere ITCZ (Fig. 3a) and the poleward flank of the southern-hemisphere ITCZ (Fig. 3c). The SPI changes arise primarily from the weakening of convection and the associated vorticity changes. More specifically, the weaker convection near 10°N and 15°S indicates less generation of low-level vorticity from the stretching of the atmosphere column. While the SPI's vorticity term is also affected by the meridional gradient of vorticity, the local changes in vorticity generally prevail over the gradient changes induced by the contraction of tropical convection. The stronger equatorial convection and weaker off-equatorial convection are also accompanied by other environmental changes. For example, the contraction of tropical convection can dry the mid-troposphere[31] (Fig. 3b) or enhance the vertical wind shear (Fig. 3d) poleward of the ITCZ, which can suppress TC development via mechanisms independent of TC "seeds".

The definition of the SPI and its intricate associations with other environmental factors warrant remarks. Specifically, the SPI is defined with a component of the vertical mass flux[50,51], which can be linked to convection changes, as well as surface fluxes and saturation deficit[49]. As suggested in the introduction paragraphs, those factors have been used to explain changes in future TC activity. One could argue that the projected changes in TC frequency may be attributed to the changes in vertical mass flux or moisture without invoking assumptions about the role of TC "seeds". While these physical hypotheses warrant future research, we speculate that the key environmental factor(s) will ultimately connect to changes in tropical convection, so our argument about the importance of the convection contraction will unlikely rest on the validity of a specific hypothesis.

To further illustrate the importance of the convection contraction at the peaks of TC seasons, we examine the relationship between the simulated changes (4 K warming - historical) in the global TC frequency and tropical convection (Fig. 4). This analysis leverages the large number of climate scenarios driven by natural and anthropogenic oceanic perturbations in the large ensemble simulation. Hence the analysis also offers an opportunity to examine the sensitivity of tropical convection and TC frequency to the patterns of surface warming.

Motivated by Fig. 3, we partition the warming-induced contraction of tropical convection into equatorial and off-equatorial components. Overall, the equatorial convection strengthens while the off-equatorial convection weakens, suggesting the projected convection contraction is qualitatively robust. Nonetheless, details of the projected contraction are dependent on the warming patterns. For example, the convection changes in the southern hemisphere are relatively weak in simulations forced by two warming patterns (CC and GF; Methods). These simulations are likely responsible for the overall muted southern-hemisphere changes projected by the large-ensemble simulations (Fig. 1b), which contrast with the projections by CMIP6 models (Supplementary Fig. 2). The inconsistency suggests larger uncertainty of our southern-hemisphere results. Compared to the other warming patterns (HA, MI, MP, and MR), the CC and GF warming patterns also show weaker El Niño-like warming in the tropical Pacific (Supplementary Fig. 5). The pattern difference likely affects regional TC activity together with changes in the South Pacific Convergence Zone[54]. Since the tropical Pacific warming profoundly affects the convection contraction[35] and may involve large model errors[55], it shall be a focus of future TC research.

Turning attention to TC frequency, the sign of peak-season changes is insensitive to the warming patterns, but the magnitude of these changes depends on these patterns and associated convection changes. In the northern hemisphere, the simulated changes in TC frequency are more closely correlated with the equatorial convection (0–7.5°N; $r = 0.35$) than with the off-equatorial convection (8–20°N; $r = 0.30$). The significant correlation between projected changes in the equatorial convection and TC frequency received limited attention from previous studies. The correlation with off-equatorial convection would strengthen if the simulations driven by the MI warming pattern are removed, though what makes the MI pattern distinct in driving TC changes is unclear. In the southern hemisphere, the correlation is weaker with the equatorial convection (0–7.5°S; $r = -0.24$) than with the off-equatorial convection (8–20°S, $r = 0.71$). The strong correlation

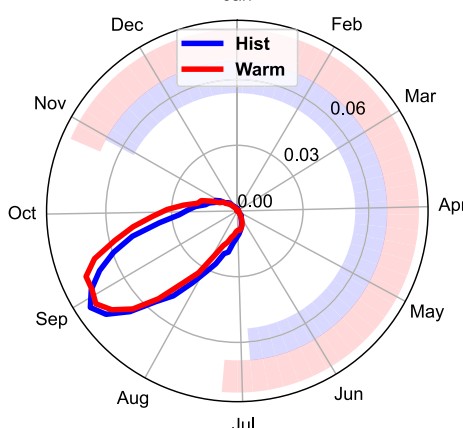

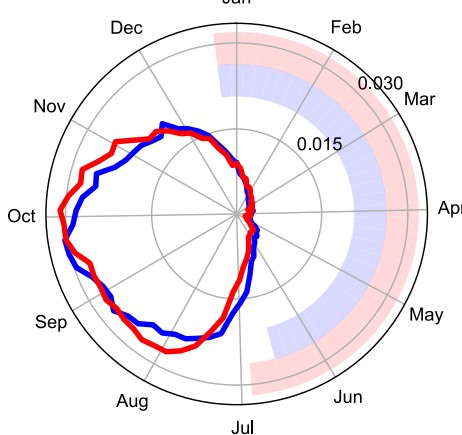

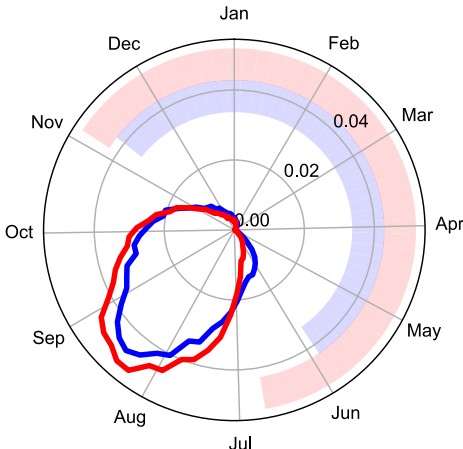

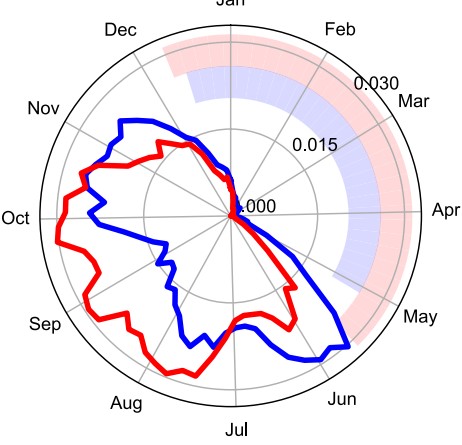

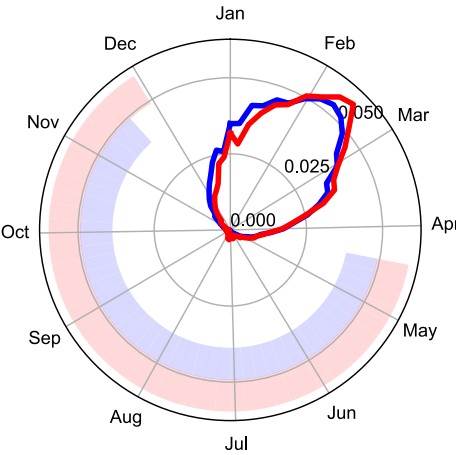

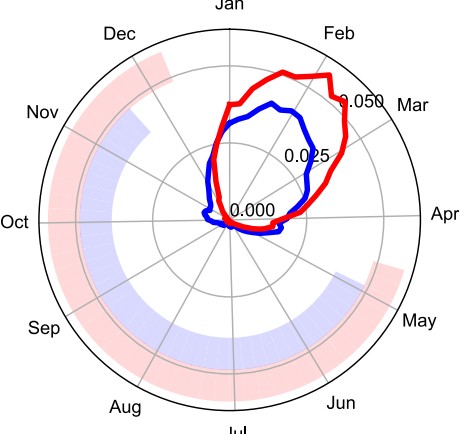

**Fig. 2 | The seasonal cycle of tropical cyclone (TC) genesis in the historical and the warming simulations.** The probability distributions of TC genesis are shown as lines for the historical (blue) and warming (red) simulations. Six ocean basins are examined: **a** North Atlantic, **b** Northwestern Pacific, **c** Northeastern Pacific, **d** North Indian Ocean, **e** South Pacific, and **f** South Indian Ocean. The polar coordinates show the time of seasonal cycle and the annual probability distributions (unitless) in angular and radial axes, respectively. The probability distributions of TC genesis are shown as lines for the historical (blue) and warming (red) simulations. The genesis time of individual TCs is grouped into 5-day bins (pentads) based on the corresponding day of the year (Methods). The mean pentads in the historical and warming simulations are denoted at the top of subplots. The outer thickened arcs indicate the periods with suppressed TC activity defined with the 0.01-probability threshold (Methods). Source data are provided as a Source Data file.

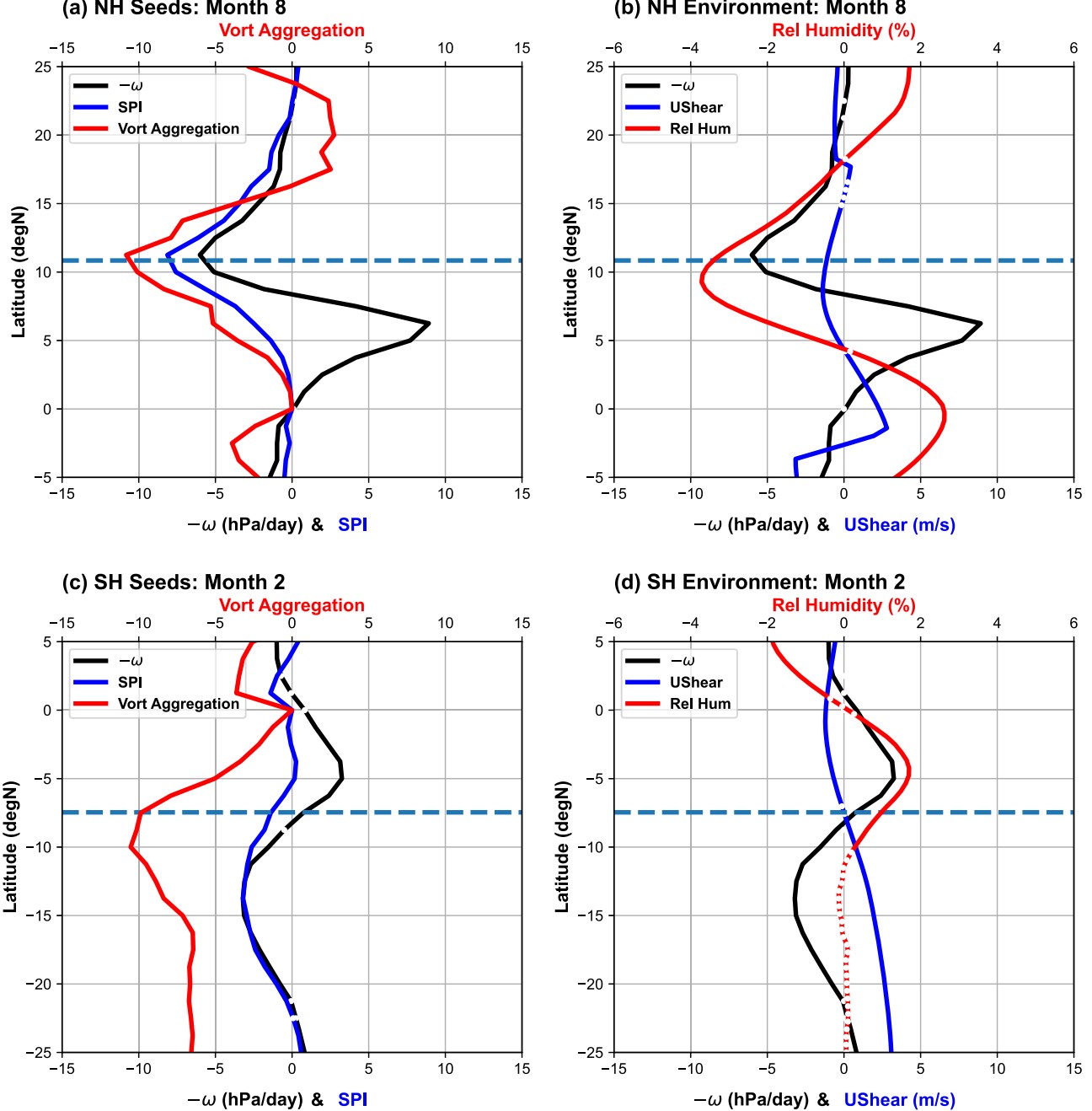

**Fig. 3 | Warming-induced changes in the latitudinal profiles of seed propensity index (SPI) and selected large-scale environmental variables.** The results near the climatology peak of tropical cyclone (TC) seasons in (**a, b**) the northern hemisphere and (**c, d**) the southern hemisphere are displayed separately. **a** August changes in the SPI and its two terms, namely the negative 500-hPa vertical motion and the 850-hPa vorticity aggregation. **b** August changes in the negative vertical motion (hPa day⁻¹), mid-troposphere (600-hPa) relative humidity (%), and vertical shear of zonal wind (m s⁻¹). (**c, d**) are the same as (**a, b**), except for February changes. The horizontal dashed line indicates the latitudes of weighted Intertropical Convergence Zone (ITCZ) of the historical simulation (HPB) in the corresponding months. The changes below the 99% confidence level are marked with white breaks on the latitudinal profile lines. Source data are provided as a Source Data file.

in the southern hemisphere (Fig. 4d) may be surprising since a large area without TC activity (180°W-30°E) could render the changes in the global zonal means less relevant. The strong correlation with off-equatorial convection is contributed by the South Indian and the South Pacific basins. In particular, the South Pacific Convergence Zone shows a noteworthy equatorward displacement at the peak TC season (Supplementary Fig. 6).

## Discussion

Recognizing the contraction of tropical convection and the inherent link between its equatorial and off-equatorial components helps identify new research opportunities. For weather-scale processes, the equatorial convection might suppress the nearby off-equatorial convection by stabilizing the troposphere and inducing downdrafts. Such a pattern of differential convective heating can either sustain a dynamically stable regime with aggregated convection and infrequent TC genesis[56] or result in dynamic instability that facilitates vortex roll-ups and periodic TC genesis[57,58]. Which scenario might prevail with anthropogenic warming warrants further investigation. From the climate perspective, the TC frequency decrease was often attributed to the weakening of tropical updraft, which has been considered as a circulation response needed to keep the global energy transport

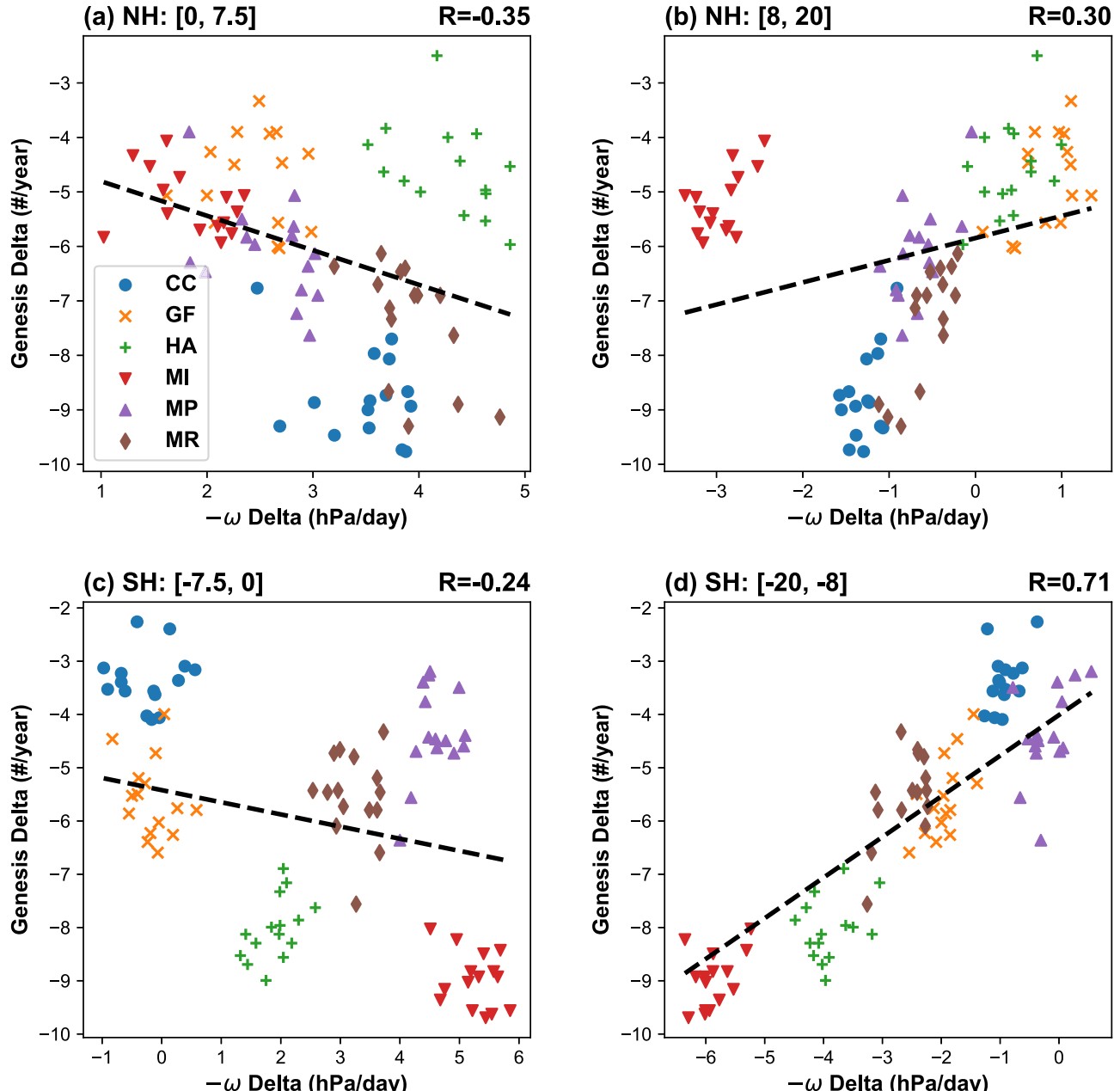

**Fig. 4 | Relationship between the warming-induced changes in the tropical convection and the global frequency of tropical cyclone (TC) genesis.** The evaluated months correspond to the peak TC seasons in (**a, b**) the northern hemisphere (August-October) and (**c, d**) the southern hemisphere (January–March). **a, c** Equatorial convection (0-7.5°°; -ω at 500 hPa) and **b, d** off-equatorial convection (8–20°) are evaluated for the hemisphere where the Inter-tropical Convergence Zone (ITCZ) and most TCs reside. The displayed changes are defined as the difference between the 4-K simulation and the ensemble mean of the historical simulations. Individual dots show ensemble members that are forced by six oceanic warming patterns (CC, GF, HA, MI, MP, and MR; see Methods). The regression of ensemble members is shown with black dashed lines. The correlation coefficients (*r*) are shown in the upper right of corresponding subplots and are all above the 95% confidence level. Source data are provided as a Source Data file.

relatively stable when anthropogenic warming moistens the atmosphere[8]. Yet a contraction of tropical convection may also serve to satisfy the same energetic constraint[59]. When compensated by stronger equatorial convection, an additional reduction in the off-equatorial convection would be possible. To what extent this compensation occurs has apparent implications for the model-projected decrease in global TC frequency.

While some other factors in our discussion (e.g., weaker mass flux, mid-tropospheric drying, and fewer tropical disturbances) have been examined separately by previous studies, this study highlights that these factors can be consolidated into a conceptual framework centered on the tropical convection. Overall, the results of this study suggest that the contraction of tropical convection, which consists of stronger equatorial convection and weaker off-equatorial convection, is important for explaining the response of TC frequency to the anthropogenic warming simulated by climate models. The contraction delays the transition of atmospheric circulation to the TC-favoring state and potentially shortens TC seasons by 5–10% with 4-K warming. The changes in the transition season on average accounts for ~75% of the TC frequency decreased projected by the large ensemble simulation. At the peak of TC seasons, the contraction suppresses the development of convective disturbances and makes the large-scale

circulation deviate from the TC-favoring state. Admittedly, the convection contraction does not explicitly address the TCs at the tropical-extratropical interface, which poses an additional threat in a warmer climate[17,18]. Nonetheless, this contraction affects the upper-level outflow of the Hadley circulation and the subtropical westerlies[31] and thus may exert an indirect influence on high-latitude TCs[18].

Our attempt to analyze more high-resolution climate simulations[60,61] was hindered by data availability issues (e.g., missing variables). We thus acknowledge that our conclusions partly depend on the fidelity of the examined models in simulating the multi-scale physical processes and their changes due to anthropogenic warming. At the same time, observational studies have found signals consistent with the simulated contraction, including a seasonal delay of tropical precipitation[62] and a contraction of the Pacific ITCZ[63]. Although this study does not seek to attribute the observed changes in global TC frequency, the observed contraction appears consistent with a declining trend in the number of TCs globally[22]. Finally, the convection contraction explains a substantial portion of the uncertainty in the projection of global TC frequency by the large-ensemble climate simulations (Fig. 4). Further coordinated efforts in theoretical development, climate modeling, and careful consideration of additional climate processes (e.g., the planetary waves and midlatitude circulations) will increase confidence in projections of future TC risk.

## Methods

### Observational data
The reference climate data in this study is the observation-constrained ERA5 reanalysis[64]. We use the monthly data on a 0.25-degree grid, which represent the tropical convection at a spatial resolution much higher than most earlier reanalysis datasets. The observational data of tropical cyclones (TCs) is the International Best Track Archive for Climate Stewardship (IBTrACS)[65]. For the sake of a relatively homogeneous definition of TC genesis across the global basins, we use the US-sourced subset of storm data in the IBTrACS. The historical analyses focus on the 30-year period of 1981–2010, which is covered by relatively reliable satellite observations and simulated by the most recent climate models.

### Coupled Model Intercomparison Project (CMIP) Data
CMIP6 data are analyzed to illustrate the robustness of the convection traction. Our analysis uses a subset of models with high-resolution configurations (~1-deg grid) available in the CMIP6[30] archive. The selected models include BCC-CSM2-MR, CMCC-CM2-SR5, CNRM-CM6-1-HR, CSIRO-ARCCSS-CM2, EC-Earth3, GFDL-CM4, HadGEM3-GC31-MM, MIROC6, MPI-ESM1-2-HR, MRI-ESM2-0, NCAR-CESM2, and TaiESM1. To facilitate comparison with the ERA5 reanalysis and the large-ensemble simulations, we use the Atmospheric Model Intercomparison Project (AMIP) simulations as the reference climate and the high-end Shared Socioeconomic Pathway (SSP5-85) simulations as the warming climate. Moreover, the strong forcing helps generate signals that are relatively easy to identify with single runs of climate models. This study also uses the CMIP5[66] data indirectly as their warming patterns serve as input for the large-ensemble simulations (see next section).

### High-resolution large ensemble simulation
The analyzed large-ensemble simulation is a subset of the Database for Policy Decision Making for Future Climate Change (d4PDF)[29]. The simulation is conducted using Meteorological Research Institute–Atmospheric General Circulation Model version 3.2 (MRI-AGCM3.2H)[67] on a grid with ~60-km grid spacing. The historical (1951–2010) simulation covers 1951–2010 and is driven by the observed forcings including the time-varying oceanic information. A 100-member ensemble simulation is generated by perturbing the initial conditions of the atmosphere and the sea surface temperature

(SST). To account for observational uncertainties, the input SST forcing also includes random perturbations up to 30% of the observed interannual variability of the SST.

The future warming simulation uses the greenhouse gas forcing that corresponds to the 2090 values of the CMIP5's representative concentration pathway 8.5 scenario. Its input SST forcing is based on the detrended SST observations plus representative SST warming patterns from six CMIP5 models. The models include HadGEM2-AO, MPI-ESM-MR, MRI-CGCM3, NCAR-CCSM4, GFDL-CM3, and MIROC5–and are denoted as HA, MP, MR, CC, GF, and MI in Fig. 4. In these simulations, each warming pattern is scaled to be 4-K warmer relative to the preindustrial climate and then serves as input to generate a 15-member ensemble. The SST warming patterns vary seasonally, and Supplementary Fig. 5 shows the annual means of these warming patterns. The model settings and the experiment design are described in detail by Mizuta et al.[29].

The large ensemble simulations produce a realistic historical climatology (including the large-scale environment and TC activity[68,69]) and a warming-induced reduction in the global TC frequency as most other climate models. This set of simulations consists of several unique advantages that are leveraged by this study. For example, the simulations contain exceptionally many ensemble members that are useful for accumulating TC samples, identifying warming-induced responses, and evaluating sensitivities across ensemble members. Besides helping simulate TCs, the high model resolution (for a climate model) is useful for the analysis of the convection changes near the narrow ITCZ. Finally, the model data, including TC information and key atmospheric variables, are meritoriously archived and publicly accessible.

Unless otherwise specified, all the key analyses in this study use the data of the large-ensemble simulation. The analyses focus on the 30-year climatology of the historical period (1981–2010) and 4-K warming scenario (2081–2110) to mitigate the burden of data management and avoid less reliable observational records. The warming-induced responses are defined as the 4-K warming (4-K) minus the historical (HPB) fields. The statistical significance of those responses is determined with the Student's t-test using all the samples from the 100-member historical simulation and the 90-member warming simulation.

### Definition of the ITCZ latitude
This study determines the latitude of the ITCZ by calculating the centroid location of the tropical convection. For each month, an algorithm first identifies the grid points with top quartile convection within the latitude band of 25°S–25°N. Then the algorithm calculates the mean latitude of convection weighted by the convection intensity and the area of grid points. When the threshold excludes all grid points except for the maximum value, the search algorithm is equivalent to the commonly used metric of the ITCZ latitude, namely the latitude of tropical convection's maximum. In the transition seasons, the choice of using the top-quartile convection ensures a relatively robust definition of the ITCZ latitude when the convection maximum sticks with one hemisphere despite intense convection on the other hemisphere. The use of the top-quartile convection also excludes relatively weak convection outside the ITCZ (e.g., the subtropical precipitation). The output of this algorithm is robust against small changes in the percentile threshold.

### TC tracking and genesis
Besides the IBTrACS observational data, the TC track data of the large-ensemble simulations[29] (https://doi.org/10.20783/DIAS.640) are also used. The track data is archived by the Data Integration & Analysis System (DIAS) and provided by Yoshida et al.[68]. The tracking algorithm[68,70] is summarized as follows. The tracking algorithm considers multiple detection criteria about storm extremes and structure. It searches for candidate systems that meet certain thresholds for maximum relative vorticity and wind speed at the 850-

hPa level. In addition, the candidate systems are screened based on the presence of a warm core aloft and the vertical profile of wind speed. The screening in the Northern Indian Ocean also includes a specific check that differentiates TCs from monsoon depressions. When a candidate system meets all the criteria and lasts at least 36 h, the algorithm classifies it as a TC. The detected TCs are then grouped based on genesis ocean basins. The threshold values used in the tracking process are identical to those of Murakami et al.[70]. The values were chosen based on the model and ensures that the global number of detected TC geneses in the historical simulations and the observation are comparable.

We define the TC genesis as the first point of individual TC tracks. This definition circumvents the need of defining an intensity threshold of genesis or considering intensity biases of simulated TCs. However, an exact comparison between the observed and simulated TC geneses is hindered by differences between track methods. Since the observed TCs were mostly tracked by human forecasters and researchers, the tracking process changed over years and fundamentally differs from the automatic tracking applied to the large ensemble simulation. Such issues may moderately affect the analysis that involves the genesis location.

### Analytics of the TC seasonal cycle

We analyze the TC seasonal cycle using the histograms of TC genesis time in individual basins. More specifically, we group all the simulated TCs in each TC basin for the historical and the warming experiments, respectively. In each basin, we convert the genesis time of individual TCs into the day of year and evaluate its histogram using 73 pentad (5-day) bins. Day 366 in the leap years of the 30-yr period is assigned to the first bin. To facilitate the comparison of the distribution shapes, we scale the histograms with the corresponding basin's total TC number in the historical or the warming experiments. In addition, the periods of TC suppression (Fig. 2) are defined as the pentad bins when <1% of a basin's TCs develop.

When paired with the large ensemble simulations, the analytical approach here has advantages in detecting shifts and yielding robust results. The use of pentad bins instead of monthly bins helps identify sub-monthly shifts in the TC seasons between the historical and the warming scenarios. This technique becomes feasible as the bundling of ensemble members ensures the large sample sizes of TCs. We have considered alternatives to define the TC seasons. For example, it is possible to define the TC season length using the difference between the genesis time of the first and the last TCs in the same season. But in the basins with low TC frequency, the stochastic noises and the warming-induced TC decrease can result in ill-defined TC seasons. For example, an extremely inactive season may generate zero or one TC, making the length of the TC season ambiguous.

### Seed propensity index and the large-scale environment analysis

Since the transfer and storage of the large-ensemble simulations make it computationally challenging to track rotating convective disturbances, we assume that their frequency scales with the Seed Propensity Index (SPI)[51]. The scaling relationship is valid with a few other high-resolution climate simulations when the SPI is defined as follows:

$$S = -\omega \frac{1}{1 + Z^{-1/\alpha}} \tag{1}$$

Where $\omega$ is the mean of isobaric vertical velocity at the 500-hPa level. The following term is referred to as the vorticity aggregation term in Fig. 4 and scales with $Z^{1/\alpha}$ near the equator. Z is defined as:

$$Z = \frac{f + \zeta}{\sqrt{|\beta + \partial_y \zeta| U}} \tag{2}$$

Where $f$ and $\beta$ are the Coriolis parameter and its meridional gradients, and $\zeta$ represents the climatology relative vorticity at the 850-hPa level. The remaining parameters adopt the empirical fitting values $U = 20 \; ms^{-1}$ and $\alpha = 0.69$[51].

The original SPI definition uses zero values in regions with climatological subsidence (i.e., $\omega > 0$), but this could complicate the calculation and comparison of zonal means in different scenarios. For example, the regions with climatological convection can shift or change their size, making it difficult to directly compare the zonal means of SPI between different climate scenarios. To simplify the comparison, we calculated the zonal means without using assigned zero values. This simplification mainly affects the subtropical regions with large zonal variations in the convection distribution. Near the equator, which is at the center of our discussion, the impact of this simplification is relatively small due to the widespread footprints of convection.

The analysis of the large-scale environment uses simple metrics, the vertical wind shear of zonal wind (200–850 hPa) and the 600-hPa relative humidity, instead of the ventilation index as Hsieh et al. This choice is a practical compromise related to the computational burden of calculating the ventilation index on a 60-km grid for nearly 6000 simulation years. To conduct the calculation, the volume of input data will likely exceed 100 TB due to the involvement of vertical integrals and multiple physical variables. For our discussion of the global zonal means, we expect the results from the simple metrics to be qualitatively consistent with the results from the ventilation index.

## Data availability

Source data are provided with this paper. The intermediate climate and TC data generated in this study have been deposited in the Zenodo database (https://zenodo.org/record/8293111). The raw model data and observational data are available from third-party providers under research use licenses. The large ensemble climate dataset is available at the Data Integration and Analysis System (DIAS) (https://search.diasjp.net/en/dataset/d4PDF_GCM). DIAS also provides access to the tropical cyclone tracks (https://search.diasjp.net/en/dataset/d4PDF_tropical_cyclone). The IBTrACS dataset is available at the National Centers for Environmental Information (https://www.ncei.noaa.gov/products/international-best-track-archive). The ERA5 reanalysis is available at the National Center for Atmospheric Research (NCAR) Research Data Archive (https://rda.ucar.edu/datasets/ds633-0/). The CMIP data is available at the NCAR analysis platform (https://www2.cisl.ucar.edu/computing-data/data/cmip-analysis-platform). Source data are provided with this paper.

## Code availability

The code used to generate Figs. 1–4 is available at Zenodo (https://zenodo.org/record/8293111). Additional analysis code is available from G.Z. upon request.

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

## Acknowledgements

The authors thank the science programs (SOUSEI, TOUGOU, SI-CAT, DIAS) of the Ministry of Education, Culture, Sports, Science and Technology (MEXT) of Japan for sharing the large ensemble simulation data and the tropical cyclone data (https://doi.org/10.20783/DIAS.640). We also would like to acknowledge the data access and computing support provided by the NCAR CMIP Analysis Platform (https://doi.org/10.5065/D60R9MSP). Early discussions with Isaac Held and Adam Sobel helped motivate this study. G.Z. thanks Suzana Camargo, Tsung-Lin Hsieh, Thomas Knutson, Hiroyuki Murakami, and Yi Zhang for stimulating discussions that helped improve the study and its presentation. G.Z. also appreciates the support of the faculty start-up funds provided by the University of Illinois at Urbana-Champaign.

## Author contributions

This study was conceived by G.Z. G.Z. acquired the data and performed the data analysis. G.Z. interpreted the results and wrote the manuscript.

## Competing interests

The author declares no competing interests.

## Inclusion and ethics

All the research data are publicly available, and the data sources have been acknowledged. All the individuals whose contributions do not meet the authorship criteria have been acknowledged.
