## [Peer Review File · Nature Communications]

REVIEWER COMMENTS

Reviewer #1 (Remarks to the Author):

This paper seeks to elucidate the connection between changes in tropical cyclone (TC) frequency and the contraction of tropical convection towards the equator, particularly during the late summer months. The argument seems to be that the annual-mean contraction of tropical convection manifests as a more limited seasonal migration of the ITCZ, with off-equatorial convection decreasing at the expense of increased equatorial convection, as measured by large-scale vertical velocity fields. This then leads to reduced TC frequency, presumably because of the weaker coriolis force closer to the equator.

The ideas in this paper feel important and insightful, and they usefully connect the dots between a few important topics in climate science. At the same time, there are some important gaps in the logic and presentation which should be addressed for this paper to realize its full potential, which I detail below. If these are addressed, however, this paper should be suitable for publication in Nature Communications.

Nadir Jeevanjee

Major Comments

1. I think there is a misperception, arising from annual-mean thinking, that the ITCZ is not relevant for TC genesis because in the annual mean it is close to the equator, where the Coriolis parameter f is low. Your seasonally-resolved Figure 1 tells a different story. If I have this right it would be worth dispelling this misconception very explicitly, both in the text and in figures. For instance, you could add a panel to Fig. 1 showing zonal mean, annual-mean line plots of rainfall and TC genesis, which will maximize at different latitudes and suggest there is not a strong connection between the ITCZ and TC genesis. Then you can show the seasonally-resolved panels you already have, which show that in fact there is a strong connection, which is only present seasonally because TCs are only generated seasonally.

2. I was quite confused by Fig. 2. What is the reader supposed to see here? Comparing the red and blue curves in each panel I see only a very slight delay in onset of TC season. I also see an *increase* in genesis rates over much of the season in each panel, which contradicts the arguments in the

paper. I also did not know what was meant by "mean pentad", and it was not clear to me what the units of the radial axis are (though i assume it is some kind of TC count/histogram).

3. Figure 3 has too much data and it was hard to see the forest for the trees. To me, the key point is that in panel a, there is a dipole in the $-\omega$ changes from ITCZ contraction, but no corresponding dipole in the SPI changes; there is an SPI decrease at the present-day ITCZ corresponding to the decrease in $-\omega$, but no corresponding increase at lower latitudes, presumably due to lower f values there? Do I have this right? It would be helpful if the author could spell out what is important here, in the text and in the figures, and relegate less important detail to the SI.

4. I am confused by the result in Fig. 4a. The correlation suggests that simulations with greater increase in equatorial ascent (i.e. more negative $\Delta \omega$, presumably due to greater ITCZ contraction) have *less* of a decrease in TC genesis. Doesn't this contradict the argument of the paper, which would be that greater ITCZ contraction yields more of a decrease in TC genesis due to suppressed off-equatorial convection?

Minor comments:

line 102-104: The author may be interested in this recent work on ITCZ narrowing: Ahmed, Fiaz, et al. "A process model for ITCZ narrowing under warming highlights clear-sky water vapor feedbacks and gross moist stability changes in AMIP models." *Journal of Climate* (2023): 1-46.

Fig. 1: What are the thin, horizontal dashed lines in each panel? Are they necessary? It would also help the reader tremendously to add some information to the plot itself rather than require the reader to dig through the caption. For instance, the plot label itself could indicate what is contoured and what is colored, e.g. "(a) Observation: Global ω (contoured) & Genesis (colors)". The color bar could have a label and units, e.g. "genesis (yr⁻¹)". The label in panel c appears to be incorrect. It would also be useful to add to the caption what models are being used here.

Reviewer #2 (Remarks to the Author):

This is an interesting manuscript that explores the climatic controls of tropical cyclone formation, which is an area with great uncertainty but also of great importance for understanding changes in actual risk. Previous work has focused mainly on relationships between climate change and/or vertical mass flux or mid-level drying. Here the author attempts to unify these ideas by focusing on changes in convection on seasonal and longer timescales. The manuscript makes good arguments and presents a novel view of the problem, and I would like to see it get published. I have just a few relatively minor comments that should be addressed.

As a general comment, extensive editing to improve grammar and syntax and modify several awkwardly phrased sentences would substantially improve readability.

In figure 1c, the title appears to be erroneous. Also, it needs to be explicitly specified that the shading represents the difference field.

In figure 2, “time-polar” is an incomplete descriptor. The second (radial) coordinate should be defined explicitly as providing a measure of probability. Also, rather than asking the reader to eyeball it from thick often overlapping lines, providing some actual numeric probability change values in the text might help to emphasize how large the differences can be. Casting those values as percent change will further emphasize this. Also, the behavior in the NI is worth a few words, given how exaggerated the signal is. It’s a bit off the main topic of this manuscript, but my interpretation is that in addition to the early- and late-season changes, the 4K warming world no longer sees a suppression of formation during the historical period of the monsoon. This is probably worth at least noting, and if such a change on the monsoon or a change to the effect of the monsoon on TC formation is already understood, then this should be noted.

Regarding the discussion beginning on line 181, the question of whether seed propensity can be assumed stationary is being actively debated in the literature. It’s been argued that Emanuel’s constant seeding technique is the root cause for his TC formation increasing with GHG warming, in some conflict with the many other papers cited in this manuscript. Surely this demands some focused discussion here.

Reviewer #3 (Remarks to the Author):

Review of “Warming-induced contraction of tropical convection reduces tropical cyclone frequency”.

Author: Gan Zhang

General comments:

This study investigates simulated changes of tropical convection, namely a contraction of the ITCZ, and associated changes in tropical cyclone frequency. The manuscript raises some very interesting ideas about the dependence of global and regional TC frequency on circulation changes. However, in its current form, I feel that the overarching story is not clear enough nor the main conclusions strong enough for a broad readership and may be better suited to a more technical journal. I have some major and minor comments below for the author’s consideration:

Major comments:

1) The main result of the paper is shown in Fig. 1. If I interpret Fig. 1b correctly, the salient circulation changes in terms of omega (the “contraction”) occur in the period between about April to July (i.e., during a period of relatively low global TC activity), whereas TC frequency peaks in January-March (Southern Hemisphere) and August-September (Northern Hemisphere). This temporal confinement of the enhanced near-equatorial convection to Austral autumn is even more pronounced in the rainfall changes shown in Supplementary Fig. 3.

Conversely, the peak climatological simulated TC activity in the NH, from about August to October (Fig. 1c), and the associated projected decrease in frequency, appear to be in a region of mixed sign in terms of the omega pattern and offset from the main circulation changes. Therefore, it is somewhat unclear what the impact of near-equatorial convection is on TC frequency during this peak period.

In the Southern Hemisphere (SH), the stronger equatorial convection / ITCZ is well-removed from the region of decreased TC activity further poleward, and the simulated omega changes are lower in magnitude, so it is unclear what the physical mechanism is linking SH TC genesis to the contraction of convection. Indeed, Fig. 4c indicates close to zero correlation between SH TC genesis change and

equatorial convection change. Most climate models exhibit a more robust decrease in SH TC frequency with warming compared to the NH (e.g., Knutson et al. 2020), so it is important that any circulation change or theory be able to explain this decrease also. Although the large-scale environmental changes are qualitatively supportive of reduced TC frequency in the SH (Fig. 3 c, d), the absence of a relationship between the strengthening of equatorial convection and SH TC frequency is a current weakness of the study.

2) The simulated contraction of tropical convection is conceivably related to an El Niño-like projected state of the tropical Pacific. Although the SST patterns used to force the model are not shown, inspection of the spatial patterns in Fig. 2 of Mizuta et al. (2017) suggests that this is likely the case. Given ongoing debate around climate models' ability to correctly predict the equatorial Pacific response to greenhouse warming (e.g., Seager et al. 2019; Adam Sobel, personal communication), and the implications for TC frequency, I think some discussion around the SST warming patterns used for forcing the experiments in this study is important to include.

Minor comments:

1) The title of the paper may be overstated, given that the warming-induced contraction seems to only explain TC frequency changes in certain months of the year rather than in the core months of TC activity.

2) Abstract, line 33: the suppression of TC-seeding convective disturbances has not been demonstrated explicitly in this study; rather the author investigates how the SPI responds to changes in the large-scale environment. This distinction should be made clear in the abstract (i.e., it is the index itself that shows a decrease, rather than the convective disturbances).

3) Line 46: Some recent high-resolution GCMs also indicate increased TC frequency (e.g., Bhatia et al. 2018; Vecchi et al. 2019) so suggest citing these studies in this discussion also.

4) Line 53: Suggest rephrasing, "...development and are framed around..."

5) Line 92: If the "climate simulations" are referring specifically to the current analysis, then this should be made clear. Otherwise, please provide some references for this statement.

6) Line 121: Which level is used to compute the vertical motion shown in Fig. 1? Please include this somewhere in the caption. Also, the caption should include a description of the shading in panels b and c.

7) Lines 148-149: Can the author quantify the effect of 5-10% TC season shortening on actual TC numbers? This is implied (e.g., “a longer period of suppression suggests a reduction in annual TC frequency) but it would be useful to know what effect the shortening has on frequency, especially given most TCs occur later in the season and there may be some compensation at other times.

8) Figure 2: Please include the unit of distance from the centre in the figure caption.

9) Lines 181- 197: The assumption throughout this section is that the climatology of TC genesis is dependent on the climatology of TC seeds. Given that this relationship has not been explicitly demonstrated (i.e., it is possible that the climatology of seeding disturbances and TCs are somewhat independent in these simulations), and the ongoing debate around this issue (e.g., Emanuel 2022), the distinction between the SPI and actual model-generated seeds (which have not been analyzed) should be made clear.

10) Line 188: The pronounced decrease near the ITCZ is only evident in the Northern Hemisphere so this needs to be clarified.

11) Figure 4 and relevant discussion: The off-equatorial convection changes in the Southern Hemisphere may be partly due to changes in the South Pacific Convergence Zone (SPCZ); an important feature for TC formation in the South Pacific (e.g., Vincent et al. 2011). It would be worth mentioning this circulation feature here as a possible mechanism for off-equatorial TC frequency changes in the SH.

12) Lines 277-284: How is the observed TC genesis defined? Is it the first point along the track or based on some wind speed threshold (e.g., 35 knots)? The definition will make some difference to the latitude of TC formation so important to clarify.

13) Methods section: How are the TCs detected and tracked within the ensembles? Some description on tracking methods and statistics of TCs in the historical and future warming simulations would be helpful.

14) Methods section: A figure showing the SST warming patterns would be useful to include (perhaps as an additional supplementary figure), given that the contraction of convection largely depends on these warming patterns and the uncertainty around the equatorial Pacific SST response to GHG forcing, as noted in Major comment #2.

References:

Bhatia, K., G. Vecchi, H. Murakami, S. Underwood, and J. Kossin, 2018: Projected Response of Tropical Cyclone Intensity and Intensification in a Global Climate Model. *J. Climate*, 31, 8281–8303, <https://doi.org/10.1175/JCLI-D-17-0898.1>.

Seager, R., Cane, M., Henderson, N. et al. Strengthening tropical Pacific zonal sea surface temperature gradient consistent with rising greenhouse gases. *Nat. Clim. Chang.* 9, 517–522 (2019). <https://doi.org/10.1038/s41558-019-0505-x>

The author thanks the three reviewers for their helpful comments and suggestions. The input has motivated extensive revisions, including the updates of some figures and captions that caused confusion among reviewers. Please find point-to-point replies as follows.

Reviewer #1 (Remarks to the Author):

This paper seeks to elucidate the connection between changes in tropical cyclone (TC) frequency and the contraction of tropical convection towards the equator, particularly during the late summer months. The argument seems to be that the annual-mean contraction of tropical convection manifests as a more limited seasonal migration of the ITCZ, with off-equatorial convection decreasing at the expense of increased equatorial convection, as measured by large-scale vertical velocity fields. This then leads to reduced TC frequency, presumably because of the weaker coriolis force closer to the equator.

The ideas in this paper feel important and insightful, and they usefully connect the dots between a few important topics in climate science. At the same time, there are some important gaps in the logic and presentation which should be addressed for this paper to realize its full potential, which I detail below. If these are addressed, however, this paper should be suitable for publication in Nature Communications.

Nadir Jeevanjee

Major Comments

1. I think there is a misperception, arising from annual-mean thinking, that the ITCZ is not relevant for TC genesis because in the annual mean it is close to the equator, where the Coriolis parameter f is low. Your seasonally-resolved Figure 1 tells a different story. If I have this right it would be worth dispelling this misconception very explicitly, both in the text and in figures. For instance, you could add a panel to Fig. 1 showing zonal mean, annual-mean line plots of rainfall and TC genesis, which will maximize at different latitudes and suggest there is not a strong connection between the ITCZ and TC genesis. Then you can show the seasonally-resolved panels you already have, which show that in fact there is a strong connection, which is only present seasonally because TCs are only generated seasonally.

The author appreciates the suggestion and has added zonal mean, annual mean panels to the right of Figure 1. The plot suggests the mean convection has an off-equator peak that is away from the peaks of TC activity. Accordingly, the updated text inserted a note: “While the annual means of TC activity and tropical convection do not share the same latitudes, their seasonal cycles suggest a close association (Figure 1a).” Then the discussion went on to discuss the seasonal dependence.

This note is kept brief as TC researchers appear aware of the seasonal dependence of TC activity on tropical convection, as suggested by studies of the Pacific ITCZ (e.g., Wang and Magnusdottir 2006) and West Africa monsoon (e.g., Gray 1990). A latter paragraph also

discussed the value of the seasonally resolved analysis (“The robustly projected contraction of tropical convection has so far escaped the scrutiny of many TC-climate studies...”)

Finally, reviewer’s comment also motivated the author to include a panel showing the seasonal cycles of meridional aggregations (Fig. 1c bottom). It provides additional information of the seasonal climatology and changes of TC activity, thus helping the transition to the discussion of Figure 2.

References

- Wang, C., and G. Magnusdottir, 2006: The ITCZ in the Central and Eastern Pacific on Synoptic Time Scales. *Mon. Wea. Rev.*, 134, 1405–1421.
- W. M. Gray, 1990: Strong Association Between West African Rainfall and U.S. Landfall of Intense Hurricanes. *Science*, 249,1251-1256.

2. I was quite confused by Fig. 2. What is the reader supposed to see here? Comparing the red and blue curves in each panel I see only a very slight delay in onset of TC season. I also see an *increase* in genesis rates over much of the season in each panel, which contradicts the arguments in the paper. I also did not know what was meant by "mean pentad", and it was not clear to me what the units of the radial axis are (though i assume it is some kind of TC count/histogram).

The author apologizes for the confusion. The update text now states that the radial axis indicates the probability distributions instead of histograms. The probability distributions are unitless and help compare the seasonal phase between historical and warming simulations. The changes in the absolute values of TC frequency are available in supplementary Figure 4. As expected by the reviewer, the absolute changes are negative across the six examined basins. The discussion and figure caption have been updated to clarify those details.

The updated caption also briefly introduced the pentad: “The genesis time of individual TCs is grouped into 5-day bins (pentads) based on the corresponding day of the year. The mean pentads in the historical and warming simulations are denoted at the top of subplots.” More information about the pentad analysis is available in the “Analytics of the TC Seasonal Cycle” section of the Methods.

3. Figure 3 has too much data and it was hard to see the forest for the trees. To me, the key point is that in panel a, there is a dipole in the $-\omega$ changes from ITCZ contraction, but no corresponding dipole in the SPI changes; there is an SPI decrease at the present-day ITCZ corresponding to the decrease in $-\omega$, but no corresponding increase at lower latitudes, presumably due to lower f values there? Do I have this right? It would be helpful if the author

could spell out what is important here, in the text and in the figures, and relegate less important detail to the SI.

The author agrees that the original discussion of Figure 3 was not strong. All the three reviewers identified gaps from different angles, so the author rewrote the discussion about Figure 3 and SPI. The revised text framed the discussion in a bigger context.

The discussion has been extended to state key assumptions and alternative interpretations. The discussion of Figure 3 also stated its aim more clearly, namely serving as an example of linking existing hypotheses about future TC changes to the contraction of tropical convection. This is the essence of Figure 3 and its discussion.

The author considered relegating some variables in Figure 3. However, such changes would make it hard to make the discussion balanced amid the ongoing debate about the role of TC “seeds”. As suggested by the revised text, the original interpretation of the TC changes with the SPI involves several assumptions. Meanwhile, the SPI has intricate associations with other environmental variables. One might argue that the TC changes could be interpreted without invoking “seeds” and fully based on environmental changes such as shear and humidity (Figs. 3b and 3d). As a result, it is hard to keep the SPI changes at the center of this discussion and relegate other variables.

Regarding the SPI detail that intrigued Reviewer #1, the lack of an SPI increase in the equatorial region appears related to changes in relative vorticity. Specifically, a decrease in the meridional gradient of relative vorticity compensates changes caused by the omega term. This issue, together with the differential convective heating (see “For weather-scale processes...” comment in the main text), warrant a numeric study in the future.

4. I am confused by the result in Fig. 4a. The correlation suggests that simulations with greater increase in equatorial ascent (i.e. more negative $\Delta \omega$, presumably due to greater ITCZ contraction) have *less* of a decrease in TC genesis. Doesn't this contradict the argument of the paper, which would be that greater ITCZ contraction yields more of a decrease in TC genesis due to suppressed off-equatorial convection?

Thanks for pointing out this issue. It is likely an artifact related to the extended temporal range of the analysis. The convection changes are the most pronounced around May but most TCs occur after August. The temporal mismatch of the original figure made it difficult interpret the results.

The author has updated the plot to focus on the peak TC months, namely Aug-Oct for the NH and Jan-Mar for the SH. This mitigates the temporal mismatch while addressing a Major comment by Reviewer #3. Additionally, the new plot switches from omega to $-\omega$ to make the treatment of vertical motion consistent across all the plots discussed by the main text.

The new Figure 4 now shows the expected correlations. More equatorial convection corresponds to fewer TCs, while more off-equator convection corresponds to more TCs. The correlations are statistically significant and consistent with the expectation that the contraction suppresses TC activity. Finally, the revised text also emphasized that both the equatorial and off-equator changes were considered as part of the convection contraction.

Minor comments:

line 102-104: The author may be interested in this recent work on ITCZ narrowing: Ahmed, Fiaz, et al. "A process model for ITCZ narrowing under warming highlights clear-sky water vapor feedbacks and gross moist stability changes in AMIP models." *Journal of Climate* (2023): 1-46.

The work is indeed interesting and relevant. It was likely in review when this study was submitted. Based on that work and references therein, the description of the physical mechanisms has been revised. A new comment on the spread of the projected contraction is also added: "The magnitude of the projected contraction depends on warming patterns (Zhou et al. 2019), as well as model details of moist static energy and vertical velocity profiles (Ahmed et al. 2023)".

Fig. 1: What are the thin, horizontal dashed lines in each panel? Are they necessary? It would also help the reader tremendously to add some information to the plot itself rather than require the reader to dig through the caption. For instance, the plot label itself could indicate what is contoured and what is colored, e.g. "(a) Observation: Global ω (contoured) & Genesis (colors)". The color bar could have a label and units, e.g. "genesis (yr-1)". The label in panel c appears to be incorrect. It would also be useful to add to the caption what models are being used here.

The horizontal dashed lines have been removed. The author also agrees with the other suggestions and have implemented them accordingly.

Reviewer #2 (Remarks to the Author):

This is an interesting manuscript that explores the climatic controls of tropical cyclone formation, which is an area with great uncertainty but also of great importance for understanding changes in actual risk. Previous work has focused mainly on relationships between climate change and/or vertical mass flux or mid-level drying. Here the author attempts to unify these ideas by focusing on changes in convection on seasonal and longer timescales. The manuscript makes good arguments and presents a novel view of the problem, and I would like to see it get published. I have just a few relatively minor comments that should be addressed.

As a general comment, extensive editing to improve grammar and syntax and modify several awkwardly phrased sentences would substantially improve readability.

The author appreciates the comment and has made extensive effort to improve the readability of this manuscript (e.g., revising figures and captions, as well as the main text). These updates are visible in the manuscript file with tracked changes. The author would welcome any additional suggestions.

In figure 1c, the title appears to be erroneous. Also, it needs to be explicitly specified that the shading represents the difference field.

The title of Figure 1c has been corrected. The new titles and the figure caption have explicitly specified that the shading represents differences between the warming and the historical simulations.

In figure 2, “time-polar” is an incomplete descriptor. The second (radial) coordinate should be defined explicitly as providing a measure of probability. Also, rather than asking the reader to eyeball it from thick often overlapping lines, providing some actual numeric probability change values in the text might help to emphasize how large the differences can be. Casting those values as percent change will further emphasize this. Also, the behavior in the NI is worth a few words, given how exaggerated the signal is. It’s a bit off the main topic of this manuscript, but my interpretation is that in addition to the early- and late-season changes, the 4K warming world no longer sees a suppression of formation during the historical period of the monsoon. This is probably worth at least noting, and if such a change on the monsoon or a change to the effect of the monsoon on TC formation is already understood, then this should be noted.

The author agrees that the caption and discussion of Figure 2 had many issues. The description has been rewritten based on the reviewer’s input.

The revised text stated that “the probability distributions of TC genesis in the historical and warming simulations show differences with magnitudes up to 0.15”. While it is logically awkward to describe fractional changes in probabilities, the author took a different

approach and revised Supplementary Figure 4 to include absolute value changes in four periods of three months. This allows a statement as follows: “the transition periods account for ~13% (North Atlantic) to ~200% (East Pacific) of the projected decreases in TC frequency, with the six-basin average being ~75%.”

The author also agrees with the reviewer’s characterization of the projected changes in the Northern Indian (NI) basin. The revised text now includes a note based on three relevant studies (“Such seasonality changes are pronounced in the Northern Indian basin ...”). The author’s reasoning basis is as follows:

- The NI changes are broadly consistent with findings by Murakami et al. (2013), which reported a robust reduction of TC activity during the pre-monsoon season and an increase during the peak-monsoon season. Their simulations were generated with two versions of the MRI AGCM, including one that was used to generate the large ensemble simulations analyzed by this study.
- Bell et al. (2020) recently used a different TC tracking algorithm and examined more models. They also found a robust reduction of pre-monsoon TC activity (their Table 3). While the peak-monsoon activity is ambiguous, the projected changes are either an increase (as Murakami et al. 2013) or a relatively weak decrease.
- Meanwhile, CMIP models suggest the Indian monsoon may experience changes including a delay of maximum precipitation (or phase shift) and an intensification of peak-season precipitation (Bombardi and Boos 2021).

References

- Murakami, H., Sugi, M. & Kitoh, A. Future changes in tropical cyclone activity in the North Indian Ocean projected by high-resolution MRI-AGCMs. *Clim Dyn* 40, 1949–1968 (2013).
- Bell, SS, Chand, SS, Tory, KJ, Ye, H, Turville, C. North Indian Ocean tropical cyclone activity in CMIP5 experiments: Future projections using a model-independent detection and tracking scheme. *Int J Climatol*. 2020; 40: 6492– 6505.
- Bombardi, R. J., and W. R. Boos, 2021: Explaining Globally Inhomogeneous Future Changes in Monsoons Using Simple Moist Energy Diagnostics. *J. Climate*, 34, 8615–8634.

Regarding the discussion beginning on line 181, the question of whether seed propensity can be assumed stationary is being actively debated in the literature. It’s been argued that Emanuel’s constant seeding technique is the root cause for his TC formation increasing with GHG warming, in some conflict with the many other papers cited in this manuscript. Surely this demands some focused discussion here.

The author agrees that the discussion of seed propensity should be extended. The author also noticed that Reviewer #3 also pointed out some weaknesses of this study's discussion (e.g., lack of explicit statement about key assumptions).

Accordingly, the revised text has noted the ongoing debate about the definition of TC seeds and their role in TC climatology ("Given the active debate about ..."). Since this study does not seek to settle the ongoing debate about TC seeds, we state that the discussion of the SPI "serves as an example of linking existing hypotheses to the contraction of tropical convection".

As required by Reviewer #3, the revised text explicitly stated some assumptions about the SPI analysis, such as that the SPI scales with the frequency of rotating convective disturbances in this large ensemble of climate simulations. Additionally, the Method section ("Seed Propensity Index and the Large-scale Environment Analysis") was also revised to explain the motivation of using the SPI instead of directly tracking TC seeds (i.e., computational challenges related to data transfer and storage).

To make the reasoning around the SPI relatively balanced, a new paragraph discusses some alternative interpretations of the SPI-related findings ("The definition of the SPI and its intricate associations ..."). Specifically, the SPI includes a component of the vertical mass flux (Hsieh et al. 2020, 2022), which can be linked to surface fluxes and saturation deficit with the weak temperature gradient assumption (Emanuel 2022). Some could thus argue that a correlation between changes in the SPI and TC frequency may not necessarily involve TC seeds. The revised text commented on this alternative interpretation.

The author also speculates that the key environment factor(s) identified by future research of physical processes can be ultimately connected to changes in tropical convection. Therefore, the argument about the importance of the convection contraction unlikely rests on the validity of a specific seed-related hypothesis.

Reviewer #3 (Remarks to the Author):

Review of “Warming-induced contraction of tropical convection reduces tropical cyclone frequency”.

Author: Gan Zhang

General comments:

This study investigates simulated changes of tropical convection, namely a contraction of the ITCZ, and associated changes in tropical cyclone frequency. The manuscript raises some very interesting ideas about the dependence of global and regional TC frequency on circulation changes. However, in its current form, I feel that the overarching story is not clear enough nor the main conclusions strong enough for a broad readership and may be better suited to a more technical journal. I have some major and minor comments below for the author’s consideration:

Major comments:

1) The main result of the paper is shown in Fig. 1. If I interpret Fig. 1b correctly, the salient circulation changes in terms of omega (the “contraction”) occur in the period between about April to July (i.e., during a period of relatively low global TC activity), whereas TC frequency peaks in January-March (Southern Hemisphere) and August-September (Northern Hemisphere). This temporal confinement of the enhanced near-equatorial convection to Austral autumn is even more pronounced in the rainfall changes shown in Supplementary Fig. 3.

The author agrees that the most robust contraction occurs in April-July. However, a close inspection of Figure 2b also suggests the contraction appears in other months, including the peak-season months of the northern hemisphere. This can be confirmed with the time slices of two peak-season months (Figure 3), which show that convection strengthens in the equatorial region and weakens near the ITCZ axis (NH) or on its poleward flank (SH). Such dipole changes can be characterized as a contraction of tropical convection.

Conversely, the peak climatological simulated TC activity in the NH, from about August to October (Fig. 1c), and the associated projected decrease in frequency, appear to be in a region of mixed sign in terms of the omega pattern and offset from the main circulation changes. Therefore, it is somewhat unclear what the impact of near-equatorial convection is on TC frequency during this peak period.

In the Southern Hemisphere (SH), the stronger equatorial convection / ITCZ is well-removed from the region of decreased TC activity further poleward, and the simulated omega changes are lower in magnitude, so it is unclear what the physical mechanism is linking SH TC genesis to the contraction of convection. Indeed, Fig. 4c indicates close to zero correlation between SH TC genesis change and equatorial convection change. Most climate models exhibit a more

robust decrease in SH TC frequency with warming compared to the NH (e.g., Knutson et al. 2020), so it is important that any circulation change or theory be able to explain this decrease also. Although the large-scale environmental changes are qualitatively supportive of reduced TC frequency in the SH (Fig. 3 c, d), the absence of a relationship between the strengthening of equatorial convection and SH TC frequency is a current weakness of the study.

The author agrees that the convection-TC association in the peak seasons is not the clearest in Figure 1c. This was why the original Figure 3 and Figure 4 were included.

Figure 3 examined two peak months for the NH (August) and SH (February) TC activity. The original analysis examined the SPI and other large-scale environmental changes (e.g., vorticity and wind shear). The revised discussion followed the reviewer's suggestion and took a more balanced view about the SPI (reply to Minor comment #9). The revision also provided an opportunity to discuss the role of convection changes at a greater depth.

Figure 3 suggests that using 8N / 8S can partition the peak-season contraction of convection into equatorial and off-equator components. This partition is worth emphasis as the reviewer appeared to equalize the contraction to the strengthening of equatorial convection alone, whereas the author considers the weakening of off-equator convection as another important part of this contraction. This partitioning and its relevance to the convection changes have been made clearer in the revised text.

Based on this partitioning, Figure 4 examined how the equatorial and off-equator components are correlated with projected TC changes. To avoid a temporal mismatch and better focus on peak-season changes, the author shrank the temporal range of this analysis to Aug-Oct (NH) and Jan-Mar (SH), which approximately correspond to the peak seasons emphasized by the reviewer. Thanks to this change, the "close to zero correlation between SH TC genesis change and equatorial convection change" became statistically significant. Moreover, the equatorial and off-equator changes are significantly correlated with TC changes in both hemispheres. The signs of these correlations are consistent with the expectation based on our physical reasoning (i.e., more equatorial convection and less off-equator convection disfavor TC activity). Overall, the revisions better support the relevance of convection contraction signals to the model-projected TC changes.

Finally, the author concurs that the SH TC changes deserved more attention. This motivated the inclusion of several notes on the South Pacific Convergence Zone (SPCZ) (reply to Minor Comment #11, Vincent et al. 2011), as well as Supplementary Figure 6. This new figure showed the convection changes in the South Indian and South Pacific basins. An interesting result is that the SPCZ strongly contracts towards the equator in peak-season months. This contraction is also accompanied by a pronounced reduction of TCs, thus consistent with the proposed conceptual picture. Additionally, the revised text a) commented on a potential limitation of using the global zonal means to study regionally concentrated TC activity in the SH; and b) paid extra attention to the sensitivity of SH convection to warming patterns when addressing Major comment #2.

Limited by space, the manuscript could not go into greater details in some cases. The author hopes the extensive revisions help address the reviewer's concern and will welcome any additional constructive suggestions.

2) The simulated contraction of tropical convection is conceivably related to an El Niño-like projected state of the tropical Pacific. Although the SST patterns used to force the model are not shown, inspection of the spatial patterns in Fig. 2 of Mizuta et al. (2017) suggests that this is likely the case. Given ongoing debate around climate models' ability to correctly predict the equatorial Pacific response to greenhouse warming (e.g., Seager et al. 2019; Adam Sobel, personal communication), and the implications for TC frequency, I think some discussion around the SST warming patterns used for forcing the experiments in this study is important to include.

The author agrees that the simulated convection change depends on the surface warming patterns. When introducing the convection contraction, the updated text included an additional note: "The projected contraction depends on the equatorial warming (Zhou et al. 2019), as well as model details of moist static energy and vertical velocity profiles (Ahmed et al. 2023)."

The revised manuscript also extends the discussion of warming patterns when addressing the southern-hemisphere changes. Following the suggestion of Minor Comment #14, the author added Supplementary Figure 5 to show the annual means of surface warming patterns. Two warming patterns (CC and GF) show weaker El Niño-like warming in the tropical Pacific and were linked to the muted convection changes in the southern hemisphere (Figure 4c and 4d), where model disagreement is also apparent within the large ensemble (and with CMIP6 models).

Finally, the revised text also explicitly suggests the importance of studying the warming patterns: "Since the tropical Pacific warming profoundly affects the convection contraction (Zhou et al. 2019) and may involve large model errors (Seager et al. 2019), it shall be a focus of warrants of future TC research."

Minor comments:

1) The title of the paper may be overstated, given that the warming-induced contraction seems to only explain TC frequency changes in certain months of the year rather than in the core months of TC activity.

The author carefully reviewed this comment, assessed the manuscript, and considered several alternative titles. This led to a minor modification of the title to "Warming-induced Contraction of Tropical Convection Delays and Reduces Tropical Cyclones".

This new title better represents the two foci of this study, i.e., the early-season and overall length changes (Fig. 2) and the core-month changes in TC activity (Fig. 3-4). The author made additional efforts to make these two foci clearer. The revisions that address Major comment #1 also provided improved discussions of the peak-season changes and the southern hemisphere.

The author hope that the reviewer finds these revisions meaningful and help make the current title suitable for this study.

2) Abstract, line 33: the suppression of TC-seeding convective disturbances has not been demonstrated explicitly in this study; rather the author investigates how the SPI responds to changes in the large-scale environment. This distinction should be made clear in the abstract (i.e., it is the index itself that shows a decrease, rather than the convective disturbances).

The author agrees with this assessment. The revised abstract de-emphasizes the SPI and summarize the findings as “At seasonal peaks of TC activity, the equatorial and off-equator components of convection contraction are associated with environmental changes that hinder TC development”.

The description is kept short here to comply with the journal’s requirement about the abstract length. An extended and updated discussion of the SPI is available in the main text.

3) Line 46: Some recent high-resolution GCMs also indicate increased TC frequency (e.g., Bhatia et al. 2018; Vecchi et al. 2019) so suggest citing these studies in this discussion also.

Vecchi et al. (2019) has been cited in the opening paragraph of the main text. Motivated by the reviewer’s suggestion, Bhatia et al. (2018) was added to the original statement about the sensitivity to model parameters.

Furthermore, the results from these two papers were described with more details: “For example, a set of high-resolution (25-km) climate simulations suggest that future TC frequency increases (Bhatia et al. 2018) as surface warming generates more seeding convective disturbances (Vecchi et al. 2019)”.

Finally, the seed argument by Vecchi et al. (2019) was incorporated into the discussion of the ensuing paragraph: “an emerging research line suggests that TC frequency may increase due to ocean warming, which promotes convective activity (including TC-seeding disturbances [Vecchi et al. 2019]) ...”

4) Line 53: Suggest rephrasing, “...development and are framed around...”

The sentence has been updated: “One line of research development is framed around...”

5) Line 92: If the “climate simulations” are referring specifically to the current analysis, then this should be made clear. Otherwise, please provide some references for this statement.

The leading sentence attempted to summarize this study’s analysis of recent climate simulations and many past studies. This sentence has been rewritten with references as follows: “an analysis of recent climate simulations [refs] and past studies [refs] suggest that tropical convection will contract toward the equator”.

6) Line 121: Which level is used to compute the vertical motion shown in Fig. 1? Please include this somewhere in the caption. Also, the caption should include a description of the shading in panels b and c.

The caption has been revised to state that the vertical motion values are from the 500-hPa level. The shading in panels b and c corresponds to warming-induced changes has been denoted in the figure caption and subplot titles.

7) Lines 148-149: Can the author quantify the effect of 5-10% TC season shortening on actual TC numbers? This is implied (e.g., “a longer period of suppression suggests a reduction in annual TC frequency) but it would be useful to know what effect the shortening has on frequency, especially given most TCs occur later in the season and there may be some compensation at other times.

The author agrees that the effect of seasonal shortening could be complicated by compensation with the TC seasons. It may be hard to elegantly separate the effect of seasonal shortening given other factors such as phase changes in the seasonal cycle.

He we instead framed the discussion around the relative importance of transition months and peak months. Specifically, Supplementary Figure 4 has been updated to show absolute value changes in four periods that corresponds to a peak, two transitions, and a bottom in the annual cycle of TC activity. Some calculations allow a statement as follows: “the transition periods account for ~13% (North Atlantic) to ~200% (East Pacific) of the projected decreases in TC frequency, with the six-basin average being ~75%.” The results suggest the changes in the transition seasons are important.

Supplementary Figure 4 Seasonal cycle of TC activity in six basins. The TC frequency (yr⁻¹) in (a) North Atlantic, (b) Northwestern Pacific, (c) Northeastern Pacific, (d) North Indian Ocean, (e) South Pacific, and (f) South Indian Ocean is separately evaluated for the historical (blue) and the warming (red) experiments. The light and dark shading indicates the 5th-95th percentile range and the 25th-75th percentile range. The black solid line shows the observation values during 1981-2010. The changes in the annual TC frequency are denoted in the upper right of subplots. The changes aggregated over three-month periods (Jan-Mar, Apr-Jun, Jul-Sep, and Oct-Dec) are denoted at the bottom of subplots.

8) Figure 2: Please include the unit of distance from the centre in the figure caption.

The author assumes this comment is for the lines that show the probability distributions. In this case, these quantities are unitless, and this has been clarified in the revised figure caption. Other changes were also made to improve the readability of the figure caption.

9) Lines 181- 197: The assumption throughout this section is that the climatology of TC genesis is dependent on the climatology of TC seeds. Given that this relationship has not been explicitly demonstrated (i.e., it is possible that the climatology of seeding disturbances and TCs are somewhat independent in these simulations), and the ongoing debate around this issue (e.g., Emanuel 2022), the distinction between the SPI and actual model-generated seeds (which have not been analyzed) should be made clear.

The author agrees with the assessment of this comment and noticed that other reviewers also demanded an extended discussion.

The revised text has noted the ongoing debate about the definition and role of TC seeds (“Given the active debate about ...”). Since this study does not seek to settle the ongoing debate about TC seeds, the revised text also stated that the discussion of the SPI “serves as an example of linking existing hypotheses to the contraction of tropical convection”.

The author agrees that past studies (e.g., Hsieh et al. 2020; Yamada et al. 2021) haven’t fully demonstrated whether the seed frequency is correlated with TC frequency (or the SPI) in *all* the climate models - or the large-ensemble simulations examined here. Accordingly, the new text follows the reviewer’s suggestion and explicitly states the assumptions related to the SPI discussion, such as “... assumes the SPI scales with the frequency of rotating convective disturbances”. Moreover, the Method section (“Seed Propensity Index and the Large-scale Environment Analysis”) was also revised to explain the motivation of using the SPI instead of directly tracking TC seeds (i.e., computational challenges related to data transfer and storage).

To make the SPI-related discussion relatively balanced, a new paragraph was added to discuss alternative interpretations (“The definition of the SPI and its intricate associations...”). Specifically, the SPI includes a component of the vertical mass flux (Hsieh et al. 2020, 2022), which can be linked to surface fluxes and saturation deficit with the weak temperature gradient assumption (Emanuel 2022). One could argue that a correlation between changes in the SPI and TC frequency may not necessarily involve TC seeds. It could instead be interpreted as an indicator of impacts by the vertical mass flux proposed by earlier studies.

The author speculates the environment factor(s) identified by future research on TC seeds and other physical processes can be ultimately connected to changes in tropical convection. Therefore, the argument about the importance of the convection contraction unlikely rests on the validity of a specific seed-related hypothesis.

10) Line 188: The pronounced decrease near the ITCZ is only evident in the Northern Hemisphere so this needs to be clarified.

The sentence has been rewritten as “... the SPI decreases near the northern-hemisphere ITCZ (Figure 3a) and the poleward flank of the southern-hemisphere ITCZ (Figure 3c)”.

11) Figure 4 and relevant discussion: The off-equatorial convection changes in the Southern Hemisphere may be partly due to changes in the South Pacific Convergence Zone (SPCZ); an important feature for TC formation in the South Pacific (e.g., Vincent et al. 2011). It would be worth mentioning this circulation feature here as a possible mechanism for off-equatorial TC frequency changes in the SH.

The author appreciates the helpful input. The updated text has cited Vincent et al. (2011) and added a note on the SPCZ: “El Nino-like warming... likely affects regional TC activity together with changes in the South Pacific Convergence Zone (Vincent et al. 2011)”. This note is part a new extended discussion of the sensitivity of southern-hemisphere convection to the warming patterns.

The revised text also discusses the new Supplementary Figure 6, which shows the convection in the South Pacific basin contracts towards the equator, accompanied by fewer TCs in the peak season.

12) Lines 277-284: How is the observed TC genesis defined? Is it the first point along the track or based on some wind speed threshold (e.g., 35 knots)? The definition will make some difference to the latitude of TC formation so important to clarify.

The observed genesis is defined as the first point along the tracks. This definition circumvents the need to consider the intensity biases of simulated TCs. The author agrees that the genesis definition can affect the latitude-related analysis. Accordingly, an additional paragraph clarified the genesis definition and briefly discussed some of its caveats (“TC Tracking and Genesis” section in Method).

13) Methods section: How are the TCs detected and tracked within the ensembles? Some description on tracking methods and statistics of TCs in the historical and future warming simulations would be helpful.

Thanks for the comment. This study uses the TC track data (doi:10.20783/DIAS.640) archived by the Data Integration & Analysis System (DIAS). The track data is generated using the algorithm described by Murakami et al. (2012) and provided by Yoshida et al. (2017) based on the large ensemble simulation (Mizuta et al. 2017). A new Method section (“TC Tracking and Genesis”) has been added to outline the design and implementation of this tracking algorithm.

14) Methods section: A figure showing the SST warming patterns would be useful to include (perhaps as an additional supplementary figure), given that the contraction of convection largely depends on these warming patterns and the uncertainty around the equatorial Pacific SST response to GHG forcing, as noted in Major comment #2.

The author agrees that including the SST warming patterns is helpful and has added Supplementary Figure 5 to show the annual means of these patterns. This addition also helps with a more extensive discussion that addresses Major comment #2. Please see the reply to that comment for details.

References:

Bhatia, K., G. Vecchi, H. Murakami, S. Underwood, and J. Kossin, 2018: Projected Response of Tropical Cyclone Intensity and Intensification in a Global Climate Model. *J. Climate*, 31, 8281–8303, <https://doi.org/10.1175/JCLI-D-17-0898.1>.

Seager, R., Cane, M., Henderson, N. et al. Strengthening tropical Pacific zonal sea surface temperature gradient consistent with rising greenhouse gases. *Nat. Clim. Chang.*9, 517–522 (2019). <https://doi.org/10.1038/s41558-019-0505-x>

REVIEWERS' COMMENTS

Reviewer #2 (Remarks to the Author):

The author has satisfactorily addressed my comments. As a minor but important comment, I don't think that the new title is adequate because "TC delay" is ambiguous. I suggest simply adding the word "formation":

Warming-induced Contraction of Tropical Convection Delays and Reduces Tropical Cyclone Formation

Reviewer #3 (Remarks to the Author):

Review of "Warming-induced contraction of tropical convection delays and reduces tropical cyclones".

Author: Gan Zhang

Reviewer: Hamish Ramsay

I thank the author for their clarifications and thorough revisions in response to my previous review. The manuscript is now much easier to follow, and the main points are better supported by the analysis and figures. I agree with the author that I was perhaps too focused on the contraction and strengthening of the near-equatorial convection at the expense of the off-equatorial circulation changes. Figure 3 shows these circulation changes (and implications for TC formation) nicely and clears up some of the ambiguity in Fig. 1. I also appreciate the inclusion of some additional analyses on the southern hemisphere TC changes and associated circulation changes. The pronounced contraction of the ITCZ/SPCZ in the South Pacific, as shown in new Supplementary Fig. 6, is particularly intriguing and motivates future research.

Minor comments:

1) On second read, and after revisions, I now think the original title is probably fine but will leave this up to the author. The new title works well too.

2) Line 29: suggest "models have produced..."

3) Line 32: remove "the" before "tropical convection"

4) Line 35: could you be more specific here? Perhaps "delays the seasonal transition of the intertropical convergence zone and shortens..." or "delays the poleward migration of the intertropical convergence zone and shortens..." (similar to line 138) ?

5) Line 40: suggest replacing "may" with "can".

6) Line 73: suggest "levels" rather than "level".

7) Line 75: replace "in the past" with "over the past"

8) Line 86: suggest "...development and are formulated around..."

9) Lines 86-87: suggest "One line of thought is framed around..."

10) Line 88: suggest sticking to present tense, i.e., "It suggests that TC genesis is hindered..."

11) Line 89: no need for "development" here.

12) Line 90: again, suggest present tense, "suggests that the drying..."

13) Line 102: suggest, "...characterised by equatorial ascent and subtropical descent."

- 14) Line 117: remove “the” before “tropical convection”
- 15) Line 157: typo in “moisture”, “moist static energy and convection activity”.
- 16) Line 159: “a response” not “an response”.
- 17) Line 206: suggest “...and off-equatorial changes which will be discussed below.”
- 18) Line 222: perhaps include “delays” here (i.e. “...warming shortens and delays...”)
- 19) Line 231, suggest “...importance of transition periods despite some differences between basins”.
- 20) Line 240: remove “be” before “model-dependent”.
- 21) Line 288: no need for “Atlantic” to be italicised.
- 22) Line 297: remove “the” before “Emanuel’s”.
- 23) Lines 300-302: suggest rephrasing the sentence to “A detailed assessment of the differences between our large-ensemble simulations and Emanuel’s statistical-downscaling results is left for future research”.
- 24) Line 390: what is meant by “weaker” here? Weaker than what?
- 25) Line 458: “off-equatorial”
- 26) Line 465: “moistens the atmosphere”.

27) Line 488: "Our attempt to analyze more high..."

28) Lines 498-500: suggest rephrasing, "Further coordinated effort in theoretical development, climate modelling, and careful consideration of additional climate processes (e.g., planetary waves and midlatitude circulations) will lead to increased confidence in projections of future TC risk."

The author thanks the editor and two reviewers for helpful suggestions. Please find my point-to-point replies as follows.

Reviewer #2 (Remarks to the Author):

The author has satisfactorily addressed my comments. As a minor but important comment, I don't think that the new title is adequate because "TC delay" is ambiguous. I suggest simply adding the word "formation":

Warming-induced Contraction of Tropical Convection Delays and Reduces Tropical Cyclone Formation

Thanks for the suggestion. The title has been updated following the suggestion.

Reviewer #3 (Remarks to the Author):

Review of “Warming-induced contraction of tropical convection delays and reduces tropical cyclones”.

Author: Gan Zhang

Reviewer: Hamish Ramsay

I thank the author for their clarifications and thorough revisions in response to my previous review. The manuscript is now much easier to follow, and the main points are better supported by the analysis and figures. I agree with the author that I was perhaps too focused on the contraction and strengthening of the near-equatorial convection at the expense of the off-equatorial circulation changes. Figure 3 shows these circulation changes (and implications for TC formation) nicely and clears up some of the ambiguity in Fig. 1. I also appreciate the inclusion of some additional analyses on the southern hemisphere TC changes and associated circulation changes. The pronounced contraction of the ITCZ/SPCZ in the South Pacific, as shown in new Supplementary Fig. 6, is particularly intriguing and motivates future research.

Minor comments:

1) On second read, and after revisions, I now think the original title is probably fine but will leave this up to the author. The new title works well too.

Thanks. Following the suggestion by Reviewer #2, the title has been modified to “Warming-induced Contraction of Tropical Convection Delays and Reduces Tropical Cyclone Formation”.

2) Line 29: suggest “models have produced...”

Updated as suggested.

3) Line 32: remove “the” before “tropical convection”

Removed as suggested.

4) Line 35: could you be more specific here? Perhaps “delays the seasonal transition of the intertropical convergence zone and shortens...” or “delays the poleward migration of the intertropical convergence zone and shortens...” (similar to line 138) ?

The sentence has been rewritten as follows: “This contraction shortens TC seasons by delaying the poleward migration of the intertropical convergence zone”.

5) Line 40: suggest replacing “may” with “can”.

The sentence has been consolidated with the next one to meet the journal’s abstract length requirement.

6) Line 73: suggest “levels” rather than “level”.

Changed as suggested.

7) Line 75: replace “in the past” with “over the past”

Changed as suggested.

8) Line 86: suggest “...development and are formulated around...”

Changed as suggested.

9) Lines 86-87: suggest “One line of thought is framed around...”

Changed as suggested.

10) Line 88: suggest sticking to present tense, i.e., “It suggests that TC genesis is hindered...”

Changed as suggested.

11) Line 89: no need for “development” here.

Changed as suggested.

12) Line 90: again, suggest present tense, “suggests that the drying...”

Changed as suggested.

13) Line 102: suggest, “...characterised by equatorial ascent and subtropical descent.”

Changed as suggested.

14) Line 117: remove “the” before “tropical convection”

Changed as suggested.

15) Line 157: typo in “moisture”, “moist static energy and convection activity”.

The typo has been corrected.

16) Line 159: “a response” not “an response”.

Corrected as suggested.

17) Line 206: suggest “...and off-equatorial changes which will be discussed below.”

The author considered the suggestion and rewrote the sentence to make the thought flow more straightforward:

“Motivated by the season-dependent equatorial and off-equatorial changes, the ensuing discussion will examine the shortening of the TC seasons and the peak-season changes in detail.”

18) Line 222: perhaps include “delays” here (i.e. “...warming shortens and delays...”)

Changed as suggested.

19) Line 231, suggest “...importance of transition periods despite some differences between basins”.

Changed as suggested.

20) Line 240: remove “be” before “model-dependent”.

Corrected as suggested.

21) Line 288: no need for “Atlantic” to be italicised.

Changed as suggested.

22) Line 297: remove “the” before “Emanuel’s”.

Corrected as suggested.

23) Lines 300-302: suggest rephrasing the sentence to “A detailed assessment of the differences between our large-ensemble simulations and Emanuel’s statistical-downscaling results is left for future research”.

Updated as suggested.

24) Line 390: what is meant by “weaker” here? Weaker than what?

The comparison is with the other warming patterns (HA, MI, MP, and MR). This has been clarified in the updated text.

25) Line 458: “off-equatorial”

This phrase has been corrected here and elsewhere in the manuscript.

26) Line 465: “moistens the atmosphere”.

Updated as suggested.

27) Line 488: “Our attempt to analyze more high...”

Updated as suggested.

28) Lines 498-500: suggest rephrasing, “Further coordinated effort in theoretical development, climate modelling, and careful consideration of additional climate processes (e.g., planetary waves and midlatitude circulations) will lead to increased confidence in projections of future TC risk.

Updated as suggested. A minor modification is changing “will lead to increased confidence” to “... will increase confidence ...”